# Optimizing Broadband Near-Infrared Emission in Bi/Sn-Doped Aluminosilicate Glass by Modulating Glass Composition

**DOI:** 10.3390/mi13060921

**Published:** 2022-06-10

**Authors:** Song Xiang, Min Zhang, Tixian Zeng, Jiang Chen, Feiquan Zhang

**Affiliations:** 1School of Physics and Astronomy, China West Normal University, Nanchong 637002, China; xiangsong@stu.cwnu.edu.cn; 2College of Optoelectronic Engineering, Chengdu University of Information Technology, Chengdu 610225, China; zengtxnc@163.com; 3Sichuan Hetai Optical Fiber Co., Ltd., Nanchong 637002, China; chenjiang985464@163.com (J.C.); zhangfeiquan2022@163.com (F.Z.)

**Keywords:** bismuth, tin, near-infrared, silicate, glass, aluminum

## Abstract

The Bi/Sn-doped aluminosilicate glass samples were prepared using a melting–quenching method and their near-infrared (NIR) emission properties were studied. An ultra-broadband NIR emission ranging from 950 nm to 1600 nm was observed in all samples under 480 nm excitation, which covered the whole fiber low-loss window. The NIR emission spectrum showed that the maximum emission peak was about 1206 nm and the full width at half maximum (FWHM) was about 220 nm. Furthermore, the NIR emission intensity strongly depends on the composition of the glass, which can be optimized by modulating the glass composition. The Bi^0^ and Bi^+^ ions were the NIR luminescence source of the glass samples in this paper. The Bi/Sn-doped aluminosilicate glass has the potential to become a new type of core fiber material and to be applied to optical fiber amplifiers (OFAs), based on its excellent performance in ultra-broadband NIR emission.

## 1. Introduction

The NIR emission of Bi-doped silica glass in the optical communication window was discovered coincidentally in 1999 by Murata et al. [1]. Fujimoto et al. [2,3,4] Demonstrated, for the first time, the optical amplification of Bi-doped silica glass at 1.3 μm, which indicates the great potential of this mysterious material for applications in photonic devices. In 2005, Dianov et al. [5] made a breakthrough in bismuth-doped quartz fiber and successfully revealed the first continuous-wave (CW) laser from 1150 to 1300 nm. This implied that the greatest strength of Bi-doped glasses used for optical fiber amplifiers is the optical bandwidth, which cannot be achieved by any rare-earth-doped glass [6,7,8,9,10,11]. Bi-doped silica glass could be a candidate for a new fiber core material, due to its broadband NIR emission that can cover the low-loss window of quartz fiber [12]. Bi doping broadens the data transmission bandwidth of fiber-optic amplifiers and effectively improves the data transmission capability of fiber-optic systems.

A series of studies focused on the relationship between the components of the bismuth glass and the NIR luminous nature of Bi-doped silica glasses [13,14,15,16,17,18,19,20]. It was found that aluminum doping has a significant effect on the near-infrared luminescence of bismuth and the fluorescence properties of alumina-doped silica glass [13]. A series of Bi-doped germanium-borate glasses had been prepared via a melt–quenching method by Liu X et al. The dual-modulating modes of visible (380~750 nm) and near-infrared (1000~1600 nm) broadband photoemission were effectively controlled with the flexible excitation scheme [14]. Bi-doped nitrogen-germanium acid glass fibers were drained at 900 °C, which increased the spectral bandwidth of Bi-doped optical emitters to >600 nm [15]. It was reported that Sn doping can improve the sensitivity, refractive index, and stability of erbium silica glass by Chiodini et al. [16]. Numerous experiments confirmed that the NIR luminous properties can be optimized by modulating the content of Bi, Al and Sn. However, the effect of Sn doping on the NIR luminescence properties of the glass is still unclear.

In this study, we selected 64SiO_2_-5Al_2_O_3_-30CaO-1Bi_2_O_3_ as the glass matrix [21,22,23,24] to investigate the NIR luminous properties of the Bi/Sn co-mixed aluminosilicate glass system, and analyzed the NIR luminous mechanism of Bi-doped glass. The glass samples were prepared using the melting–quenching method, tested, and analyzed by absorption spectra, NIR emission spectra, and Fourier transform infrared (FTIR) spectra.

## 2. Experiments

The SiO_2_-CaO-Al_2_O_3_-Bi_2_O_3_-SnO_2_ glass samples were prepared using the traditional melting–quenching method. SiO_2_, Al_2_O_3_, SnO_2_, CaCO_3_, and Bi_2_O_3_ with high purity (99.99%) were used as starting materials. The raw material was weighed according to the molar ratio, mixed in an agate mortar, then melted in a corundum crucible at 1600 °C for 1.5 h. The melts were subsequently poured onto a cold stainless steel plate, then quickly pressed using another stainless steel plate, which ensured high cooling rates and avoided the possibility of the Bi metal separating out from the glasses, and then they were annealed at 750 °C for 5 h, in order to remove internal stresses. The oval-shaped glass samples were divided into regular squares with side length of 10 mm, and were polished on both the top and bottom surfaces in preparation for optical characterization.

Absorption spectra were recorded by a UV-VIS-NIR spectrophotometer (3600plus, Shimadzu, Japan) and detected by an InGaAs detector. NIR emission spectra were measured on a steady-state fluorescence spectroscopy spectrometer (FLS980, Edinburgh Instruments, EI, Edinburgh, Britain) with 480 nm continuous wave (CW) as the excitation source, and were detected in the range 800–1700 nm. Fourier transform infrared (FTIR) spectra were collected on a Nicolet 6700 FTIR spectrometer (Nicolet, Thermo Fisher Scientific, Waltham, MA, USA) by dispersing glass powders into KBr pellets. X-ray diffraction spectroscopy of the glass samples was recorded using a TD-3500 electronic material structure analyzer at 30 kV/40 mA, Cu/Kα, 2θ range of 5–70°. Solid-state NMR measurements of the sample were conducted at 14.01 T using a JNM-ECZ600R spectrometer (JEOL RESONANCE Inc., Tokyo, Japan) operating at a 1H resonance frequency of 599.7 MHz.

## 3. Results and Discussion

The absorption spectra of (69-x) SiO_2_-xAl_2_O_3_-30CaO-1Bi_2_O_3_ (x = 5, 10, 15, 20, 25, 30) glass samples are shown in Figure 1a. Two absorption bands were observed at 480 nm and 712 nm, which are similar to the characteristic Bi-related absorption bands observed in various glasses [7,25,26]. Both of them exhibited transmittance of about 88% in the range from 800 nm to 1200 nm. The absorbance at both 480 nm and 712 nm increased with increasing Al_2_O_3_ concentration at first and then reached a maximum at about 10 mol% Al_2_O_3_. With the further increase in Al_2_O_3_ concentration, the absorption peak at 712 nm weakened and disappeared. As compared to the spectroscopic data of both Bi^0^ and Bi^+^ in previous literature [27,28], the strongest absorption at 480 nm is assigned to the transition of ^4^S_3/2_ → ^2^P_1/2_ of Bi^0^, and the absorption at 712 nm is mainly from the transition of ^3^P_0_ → ^3^ P_2_ of Bi^+^. The photoluminescence (PL) spectra of the glass samples are presented in Figure 1b. A broadband emission at about 1215 nm, with an FWHM of more than 250 nm, could be observed in the NIR PL spectra of the 10Al sample. The intensity of the NIR emission increases first and then decreases with increasing Al_2_O_3_ concentration. It is well known that Al^3+^ is usually used as a glass modifier or network modifier, which facilitates the formation of Bi NIR centers in Bi-doped silica glass [29,30]. However, the NIR emission characteristics of the glass samples are not monotonically enhanced with increasing Al_2_O_3_ concentration. The possible reason for this result is the distribution of aluminum in the glass network [30]. The network structure of the glass would be disrupted because of superfluous Al_2_O_3_ concentration. The increase in four-coordinated tetrahedron sites (AlO_4_) in the glass network will weaken the crystal field strength around the bismuth and eventually result in blue shifts in the NIR emission [31]. The dependencies of emission intensity at 1215 nm with absorbance at 480 nm on Al_2_O_3_ concentration are shown in Figure 1c. The absorbance and luminescence properties of the glass samples changed significantly with the increase in Al_2_O_3_ concentration. The variation tendency of the emission intensity agrees well with that of the absorption intensity at 480 nm. This predicated that optimizing NIR emission in Bi-doped aluminosilicate glass could be achieved by modulating Al_2_O_3_ composition in the glass. As shown in Figure 1b, when the concentration of Al_2_O_3_ reached 10 mol%, the glass sample showed the strongest NIR emission and widest luminous range, so the amount of Al_2_O_3_ was 10 mol% in the optimized glass composition. The optimized glass formulation was 59SiO_2_-10Al_2_O_3_-30CaO-1Bi_2_O_3_. The appearance of glass samples was affected by Al_2_O_3_ concentration, and the color gradually became lighter than pink as the Al_2_O_3_ concentration increased, as shown in Figure 1d.

The NIR PL spectra of (60-x) SiO_2_-10Al_2_O_3_-30CaO-xBi_2_O_3_ (x = 0.1, 0.5, 1.0, 1.5, 2.0) glass samples are shown in Figure 2a. A broadband emission at about 1208 nm, with FWHM of about 222 nm, was observed in the NIR PL spectra of sample 0.5 Bi. The NIR emission intensity increases first and then decreases with the increase in Bi_2_O_3_ concentration. As observed in Figure 2b, the red shift of the emission peaks with increasing Bi_2_O_3_ content. The reason for the decline in emission at high bismuth doping levels was the typical concentration quenching effect caused by cross relaxation between the Bi^0^ ions that are closer together [32]. The dependencies of emission intensity and peak wavelength on Bi_2_O_3_ concentration are shown in Figure 2b. It is found that the NIR emission of the glass samples reaches a maximum when the Bi_2_O_3_ doping concentration is 0.5 mol%. The optimized composition for NIR emission was 59.5SiO_2_ -10Al_2_O_3_-30CaO-0.5Bi_2_O_3_.

The (59.5−x) SiO_2_-10Al_2_O_3_-30CaO-0.5Bi_2_O_3_-xSnO_2_ (x = 0.2, 0.4, 0.6, 0.8, 1.0) glass samples were prepared using the traditional melting–quenching method. Figure 3a indicates the absorption spectra of (59.5−x) SiO_2_-10Al_2_O_3_-30CaO-0.5Bi_2_O_3_-xSnO_2_ glasses. The inset shows the absorbance of glass samples in the range from 400 nm to 600 nm. One main absorption band is observed at about 480 nm. It can be noted that there is no apparent absorption observed around 800 nm, which has been frequently reported in germinate glasses doped with bismuth concentrations higher than 0.5 mol% [26,33]. The intensity of the absorption band at 480 nm is weakest when the SnO_2_ concentration reaches 0.8 mol%. There was no significant change in the transparency of the glasses.

The NIR PL spectra of (59.5−x) SiO_2_-10Al_2_O_3_-30CaO-0.5Bi_2_O_3_-xSnO_2_ glasses are shown in Figure 3b, which are measured by introducing an LD at 480 nm with a 75 W Xe900 lamp. An ultra-broadband NIR emission covering 900–1600 nm is observed in all samples. When the SnO_2_ concentration was increased to 0.8 mol%, the intensity of the NIR emission reached a minimum and then enhanced with increasing SnO_2_ concentration. We estimate that the blue shift of Bi NIR emission is due to the loss of low-valent Bi ions. The dependencies of emission intensity at 1206 nm with absorbance at 480 nm on SnO_2_ concentration are shown in Figure 3c; with the increase in SnO_2_ concentration, the absorption and luminescence properties of the glass samples indicate the same trend of change. Figure 1c and Figure 3c indicate that the absorption and luminescence properties of the Bi-doped glasses depend on the glass composition. Therefore, the optimized NIR PL nature of Bi/Sn-doped glass could be achieved by Bi/Sn co-mixed composition. The final optimized composition for NIR emission is 58.5SiO_2_–10Al_2_O_3_– 30CaO–0.5Bi_2_O_3_–1SnO_2_. Figure 3d illustrates the X-ray diffraction spectroscopy. The peak in the 25–30° range was a typical glass diffusion peak. No diffraction peak was observed, proving the amorphous state of the samples. The result of XRD indicated that the addition of SnO_2_ did not change the phase of the glass samples.

The luminescence properties of Bi-doped silica glass depend on the valence state of Bi. The valence of bismuth in glass is easily influenced by the surrounding environment, such as the concentration of non-bridging oxygens (NBOs) in the glass. The FTIR spectra can directly show the existence of NBOs. To understand the nature of the changes in NIR luminescence in glass samples, the FTIR spectra were measured. The FTIR spectra of (59.5-x) SiO_2_-10Al_2_O_3_-30CaO-0.5Bi_2_O_3_-xSnO_2_ glass samples are illustrated in Figure 4a. Five noticeable peaks of glass samples could be traced at ~1192, ~1047, ~985, ~714 and ~463 cm^−1^ in the FTIR spectra. The peaks at ~1192 and ~1047 cm^−1^ were Si-O-Si asymmetrical stretching vibrations. The asymmetric stretching vibration peak of the terminal Si-O bonds is at ~985 cm^−1^. The relative intensity of the vibration peak at ~985 cm^−1^ is decreases with the increase in SnO_2_ concentration, up to 0.8 mol%, and then increases with the further increase in SnO_2_ concentration, indicating that the amount of NBOs is changed and the glass is depolymerized. In addition, during the depolymerization of glass network, Si-O-Al bonds were preferentially broken down due to their longer bond length than Si-O-Si [22]. The Al-O stretching vibrations in isolated AlO_4_ (~714 cm^−1^) are decreased when more SnO_2_ is incorporated into the silica network. To support the result of the FTIR spectra, the ^27^Al NMR spectra of (59.5-x) SiO_2_-10Al_2_O_3_-30CaO-0.5Bi_2_O_3_-xSnO_2_ glass samples are collected, as shown in Figure 4b. Vast majority of the aluminum exists in AlO_4_ (60ppm), and the change in AlO_4_ coincides with the change in the FTIR spectra. AlO_4_ favors the existence of lower-valence-stated bismuth and enhances its NIR emission [31], and NBOs would reduce its NIR emission. As shown in Figure 3b, the NIR emission decreases at first and then increases with the concentration of SnO_2_; this is caused by the competitive effect of NBOs and AlO_4_.

There is still controversy about the NIR luminescence of Bi-doped silica glass, which has mainly been attributed to higher-valent Bi, such as Bi^5+^ [3], and lower-valent Bi, such as Bi^+^ [21] and Bi^0^ [7]. Qiu, J et al. [7] reported that when excited by 610 nm light, Bi^+^ and Bi^0^ are excited, but the ^2^P_1/2_ → ^4^S_3/2_ transition of Bi^0^ plays a dominant role, resulting in an emission band at 1210 nm. According to the research of Liu, X et al. [14], excitation at either 460 or 780 nm always produces broad emissions in the NIR region of 1000~1600 nm, which corresponds to the transition of ^2^D_3/2_ → ^4^S_3/2_ of Bi^0^ and the transition of ^3^P_1_ → ^3^P_0_ of Bi^+^, respectively. Combined with the spectral data (shown in Figure 1b and Figure 3b), the NIR luminous bands were around 1205 nm for all the glass samples. We suggested that both the Bi^0^ and Bi^+^ active centers may exist in our glass samples. Further assignments will be discussed. 

In general, the absorption edge is caused by the electronic transitions, which also reflects the optical band gap of materials. As developed by Tauc and Menth [34], the absorption coefficient *(α)* obeys the following relation for most amorphous materials:(1)$${\left(\alpha h\upsilon \right)}^{n}=B\left(h\upsilon -E_{g}\right)$$

Here, *α* is the absorption coefficient, and α can be obtained from the transmittance T and thickness d of the glass (α=−lgT/d); hυ is the incident photon energy; *B* is an energy-independent constant; *Eg* is the band gap energy; and *n* is the nature of transition. By plotting ${\left(\alpha h\upsilon \right)}^{n}$ as a function of *hν* (Tauc’s plot), one can find the optical band gap. Finally, values of *Eg* are determined from the extrapolation of the linear region of ${\left(\alpha h\upsilon \right)}^{n}$ = 0 and are listed in Table 1. It was seen that the optical band gap decreased slightly with the SnO_2_-doped glass sample, corresponding to the progressive red shift of the absorption edge [14]. We empirically attributed this decrease in the band gap (or red shift of the absorption edge) to the reduction in the relative content of Bi^0^; that is, more introduced SnO_2_ was converted into higher-valent Bi species in the glass samples, leading to a decrease in Bi^0^ absorption (shown in Figure 1a and Figure 3a). Meanwhile, the NIR luminescence intensity of the glass samples also decreased. The NIR luminescence center of Bi-doped glass stemming from low-valent Bi^0^ and Bi^+^ was confirmed.

## 4. Conclusions

To sum up, the influence of the concentration of aluminum and bismuth on the NIR emission properties of silica glass, based on the 64SiO_2_-5Al_2_O_3_-30CaO-1Bi_2_O_3_ glass composition, was studied separately, and then the effects of co-mixed Bi/Sn on the NIR luminescence properties of glass were investigated. According to the spectral data, we found that the NIR emission intensity has a certain dependence on the components of Bi/Sn-doped glasses. When the concentrations of Al_2_O_3_, Bi_2_O_3_, and SnO_2_ were 10, 0.5, and 1.0 mol%, respectively, the glass samples had the strongest NIR luminescence intensity, as shown in Figure 1b, Figure 2a and Figure 3b. A broadband emission at about 1206 nm, with FWHM of more than 220 nm, can be observed in the NIR PL spectra of the glass samples. The optimal formulation for NIR emission is 58.5SiO_2_-10Al_2_O_3_-30CaO-0.5Bi_2_O_3_-1.0SnO_2_ (mol%). It is considered that the Bi^0^ and Bi^+^ were the NIR luminescence source of the glass samples in this thesis. The study on the luminescence center of bismuth-doped glasses indicated that SnO_2_ doping reduces the low-valent Bi ion concentration, which reduces the NIR luminescence intensity of the glass. Tunable NIR emission was observed by changing the glass components. The present investigation will be of value for designing suitable glass composition.

## Figures and Tables

**Figure 1 micromachines-13-00921-f001:**
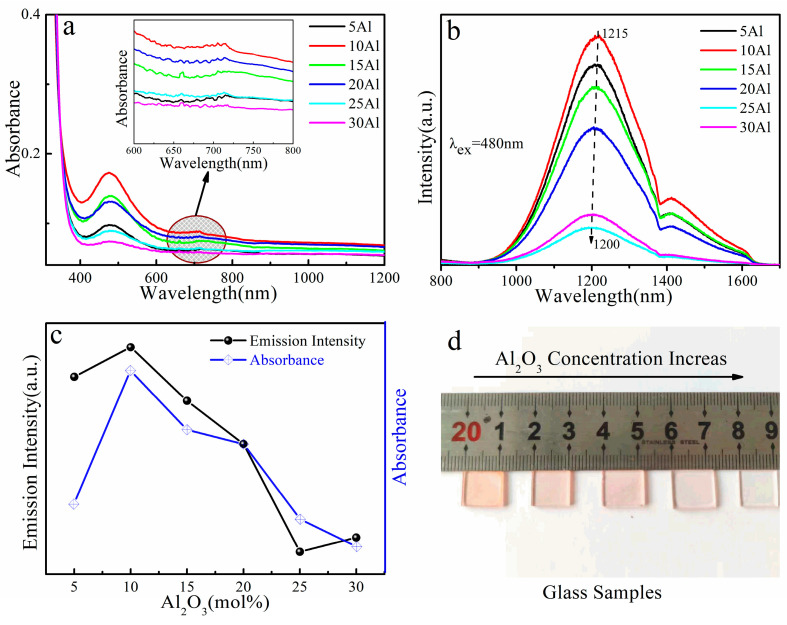
(**a**) Transmission spectra of (69-x) SiO_2_-30CaO-xAl_2_O_3_-1Bi_2_O_3_ glasses. The inset shows their transmission spectra in the range from 600 nm to 800 nm. (**b**) PL spectra of (69-x) SiO_2_-30CaO-xAl_2_O_3_-1Bi_2_O_3_ glass samples excited by a 480 nm LD. (**c**) Dependencies of emission intensity and transmittance on Al_2_O_3_ concentration. (**d**) The photo of (69-x) SiO_2_-30CaO-xAl_2_O_3_-1Bi_2_O_3_ glass samples.

**Figure 2 micromachines-13-00921-f002:**
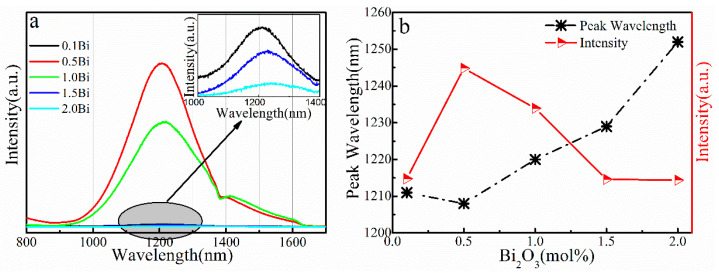
(**a**) PL spectra of glass samples excited by a 480 nm LD. (**b**) Dependencies of emission intensity and peak wavelength on Bi_2_O_3_ concentration.

**Figure 3 micromachines-13-00921-f003:**
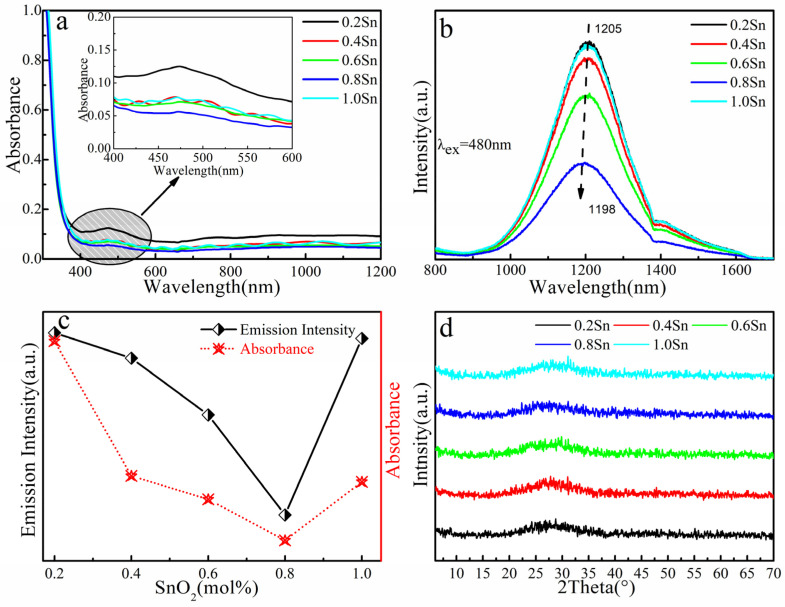
(**a**) Absorption spectra of (59.5-x) SiO_2_-10Al_2_O_3_-30CaO-0.5Bi_2_O_3_-xSnO_2_ (x = 0.2, 0.4, 0.6, 0.8, 1.0) samples. The inset shows their absorbance in the range from 400 nm to 600 nm. (**b**) PL spectra of the (59.5-x) SiO_2_-10Al_2_O_3_-30CaO-0.5Bi_2_O_3_-xSnO_2_ glass samples excited by a 480 nm LD. (**c**) Dependencies of emission intensity and absorption spectra on SnO_2_ concentration. (**d**) XRD patterns of the (59.5-x) SiO_2_-10Al_2_O_3_-30CaO-0.5Bi_2_O_3_-xSnO_2_ samples.

**Figure 4 micromachines-13-00921-f004:**
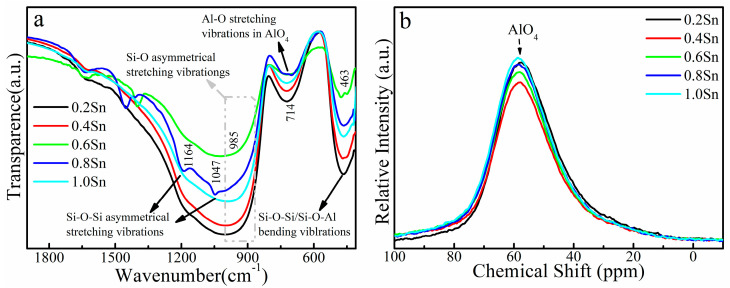
(**a**) FTIR spectra of 59.5SiO_2_-10Al_2_O_3_-30CaO-0.5Bi_2_O_3_-xSnO_2_ glass samples. (**b**) ^27^Al NMR spectra of (59.5-x) SiO_2_-10Al_2_O_3_-30CaO-0.5Bi_2_O_3_-xSnO_2_ glass samples.

**Table 1 micromachines-13-00921-t001:** Optical band gap of the glasses.

xAl Sample	Eg ± 0.01 (eV)	xSn Sample	Eg ± 0.01 (eV)
5Al	3.83	0.2Sn	3.76
10Al	3.87	0.4Sn	3.77
15Al	3.85	0.6Sn	3.79
20Al	3.84	0.8Sn	3.77
25Al	3.82	1.0Sn	3.72
30Al	3.81

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
