# Peer review of "Optimizing Broadband Near-Infrared Emission in Bi/Sn-Doped Aluminosilicate Glass by Modulating Glass Composition"

_micromachines, 2022, doi:10.3390/mi13060921_

Round 1

Reviewer 1 Report

                                   Reviewer's remarks

1. Instead of terms “quartz glass” and “silicon glass” the reviewer recommends using “silica glass”.

2. Replace CaCO3 with CaO in the formula for the glass composition.

3. Line 47-53. The last paragraph of the introduction should contain the purpose of the work but not its results.

3. Line 108. SiO2-CaCO3-xAl2O3-Bi2O3 replace with (69-x) SiO2-xAl2O3-30CaO -1Bi2O3

    Make similar corrections for glass compositions with a variable content of Bi2O3 and SnO2.

4. Line 112. (59-x) SiO2-10Al2O3-30CaCO3-xBi2O3 replace with (60-x) SiO2-10Al2O3-30CaO - Ñ…Bi2O3.

5. Line 223. 59.5SiO2 -10Al2O3 -30CaCO3 -0.5Bi2O3 -1.0SnO2 (mol%). replace with 58.5  SiO2 -10Al2O3 -30CaO  -0.5Bi2O3 -1.0SnO2 (mol%).

6. Lines 230-236. In the "Author contribution" section, authors must provide the required information. Indicate the initials of the authors instead of “XX” “XY”.

7. Remove the characters [J], [C] and [M] from references.

Author Response

Dear reviewer:

Thank you for your work and for the reviewers’ comments concerning our manuscript. The manuscript micromachines-1751105 has been carefully revised. We appreciate the detailed and useful comments and suggestions from reviewers. The point-by-point answers to the comments and suggestions were listed as below.

Reviewer #1:

  1. Instead of terms “quartz glass” and “silicon glass” the reviewer recommends using “silica glass”.

Answer: Thanks a lot for your kind suggestion. the terms “quartz glass” and “silicon glass” have modified using “silica glass” in this manuscript.

  1. Replace CaCO3with CaO in the formula for the glass composition.

Answer: The CaCO3 in the formula for glass composition has revised with CaO.

  1. Line 47-53. The last paragraph of the introduction should contain the purpose of the work but not its results.

Answer: The results have removed from the last paragraph of the introduction.

  1. Line 108. SiO2-CaCO3-xAl2O3-Bi2O3replace with (69-x) SiO2-xAl2O3-30CaO -1Bi2O3

 Make similar corrections for glass compositions with a variable content of Bi2O3 and SnO2.

Answer: We agree with this proposal that the compositions of glass has been modified.

  1. Line 112. (59-x)SiO2-10Al2O3-30CaCO3-xBi2O3 replace with (60-x) SiO2-10Al2O3-30CaO - Ñ…Bi2O3.

Answer: We agree with this proposal that the compositions of glass has been modified.

  1. Line 223. 59.5SiO2-10Al2O3-30CaCO3-0.5Bi2O3-1.0SnO2(mol%). replace with 58.5  SiO2-10Al2O3-30CaO-0.5Bi2O3-1.0SnO2 (mol%).

Answer: We agree with this proposal that the compositions of glass has been modified.

  1. Lines 230-236. In the "Author contribution" section, authors must provide the required information. Indicate the initials of the authors instead of “XX” “XY”.

Answer: We have provided the required information in the "Author contribution" section.

  1. Remove the characters [J], [C] and [M] from references.

Answer: Thanks a lot for your careful comments. The characters [J], [C] and [M] have been removed from references.

Reviewer 2 Report

In this manuscript, the Bi/Sn-doped aluminosilicate glasses showing a broadband NIR emission 950-1600 nm were prepared and the composition-dependent NIR emission were studied and discussed. The starting point of this experiment is interesting. However, the present discussion is not convincing enough, more detailed explanation or characterization must be required. In addition, the full English presentation is too poor and it needs to be improved to meet the requirements of a scientific journal publication. So, I suggest the authors make a thorough revision before submitting the Journal of Micromachines

The following comments may be useful

1.      A Nuclear magnetic resonance measurement can provide convincing evidence for the change of aluminum-oxygen tetrahedral and other polyhedron.

2.      At such a high melting temperature, aluminium diffusion from the used alumina crucible should be considered, which may affect the glass composition and finally the PL.  

3.      Why did the author choose tin oxide SnO2? What is the purpose? If the SnO (having reductive ability) was considered in your experiments

Author Response

Dear reviewer:

Thanks a lot for your kind suggestion. We have revised the whole manuscript micromachines-1751105 carefully. We appreciate the detailed and useful comments and suggestions from reviewers. The point-by-point answers to the comments and suggestions were listed as below.We hope that the language is now acceptable for further review process.  

Reviewer #2:

In this manuscript, the Bi/Sn-doped aluminosilicate glasses showing a broadband NIR emission 950-1600 nm were prepared and the composition-dependent NIR emission were studied and discussed. The starting point of this experiment is interesting. However, the present discussion is not convincing enough, more detailed explanation or characterization must be required. In addition, the full English presentation is too poor and it needs to be improved to meet the requirements of a scientific journal publication. So, I suggest the authors make a thorough revision before submitting the Journal of Micromachines.

The following comments may be useful

  1. A Nuclear magnetic resonance measurement can provide convincing evidence for the change of aluminum-oxygen tetrahedral and other polyhedron.

Answer: Thank you very much for your kind advice. The samples have been sent out for measurement. Due to the relatively short response time, the data cannot be displayed in this revision. Once measurement results come out, we will present it in the manuscript as soon as possible.

  1. At such a high melting temperature, aluminium diffusion from the used alumina crucible should be considered, which may affect the glass composition and finally the PL.

Answer: In the (69-x) SiO2-xAl2O3-30CaCO3-1Bi2O3 (x=5,10,15,20,25,30) samples, we researched the effect of the composition of alumina in glass on the PL properties, the ratio with the strongest NIR emission was selected, and the influence of the diffusion of aluminum from the alumina crucible on the sample was reduced under the same melting condition.

  1. Why did the author choose tin oxide SnO2? What is the purpose? If the SnO (having reductive ability) was considered in your experiments.

Answer: In the report of Chiodini et al [1], SnO2 doping can improve the stability and optical sensitivity of the optical fiber, which is beneficial to the preparation of the fiber. The purpose of selecting SnO2 is to explore whether Sn can broaden the near-infrared luminescence range of the fiber and enhance the luminescence. SnO begins to decompose at boiling point 1080°C, and the melting point of SnO2 is 1630°C. SnO is not suitable for use because of the high preparation temperature in this experiment. The reducibility of SnO was not considered.

[1] Chiodini, N.; Paleari, A.; Spinolo, G.; Chiasera, A.; Ferrari, M.; Brambilla, G.; Taylor, E. R. Photosensitive erbium doped tin-silicate glass. Journal of non-crystalline solids, 2002, 311(3), 217-222.

Round 2

Reviewer 2 Report

Although the english writing have improved, there are still some mistakes, such as

(1) Weighed according to the molar ratio, mixed in an agate mor-197 tar, then melted in a corundum crucible at 1600℃ for 1.5 hours.

(2) Fujimoto et al [2-4]. demonstrated 24 for the first time the optical amplification of Bi-doped silica glass at 1.3 μm...In 2005, Dianov et al [5], made a breakthrough in bismuth-doped quartz fiber and suc-27 cessfully

(3) which ensuring the high cooling rates and

...

THE AUTHORS HAVE TO TAKE THE WRITING SERIOUSLY!